# Sheetpedia: A 300K-Spreadsheet Corpus for Spreadsheet Intelligence and LLM Fine-Tuning

**Zailong Tian**[1,2]   **Zhuoheng Han**[1]   **Houfeng Wang**[1]*   **Lizi Liao**[2]

[1]State Key Laboratory for Multimedia Information Processing, Peking University
[2]School of Computing and Information Systems, Singapore Management University
wanghf@pku.edu.cn, {zltian, lzliao}@smu.edu.sg

## Abstract

Spreadsheets are widely used for data analysis and reporting, yet their complex structure and formula logic pose significant challenges for AI systems. We introduce Sheetpedia, a large-scale corpus of over 290,000 diverse spreadsheets (from 324,000+ workbooks) compiled from enterprise email archives and online forums. We detail a rigorous collection and preprocessing pipeline (integrating the Enron email spreadsheet archive and the Fuse web corpus, plus a new crawl of Excel forums) to standardize formats, filter languages, and remove duplicates. Sheetpedia provides extensive coverage of real formulas and annotations – addressing a gap left by prior table datasets (e.g. web tables used in TURL or Text-to-SQL in Spider) which often lack formula semantics. We present comprehensive corpus statistics, highlighting rich formula diversity and a majority (78%+) of English content. To demonstrate the corpus's utility, we fine-tune large language models on Sheetpedia for two novel spreadsheet understanding tasks: Natural Language to Semantic Range (NL2SR) and Natural Language to Formula (NL2Formula). Using a rejection-sampling data generation strategy, our fine-tuned models achieve up to 97.5% accuracy on NL2SR and 71.7% on NL2Formula – substantially outperforming baseline approaches. Sheetpedia (to be released publicly) fills a crucial need for a large, high-quality spreadsheet benchmark, enabling more effective spreadsheet intelligence and natural language interfaces for spreadsheet tools.

## 1 Introduction

Spreadsheets are a ubiquitous tool for data-driven decision making, used by hundreds of millions of people worldwide. They combine tabular data with formula-based computations, allowing end-users to perform complex analyses without traditional programming. However, the semi-structured nature of spreadsheets – including free-form text, numeric data, and executable formulas intermingled in cells – makes automated understanding by AI systems extremely challenging. Unlike database tables or CSV files, spreadsheets often lack a fixed schema and rely on implicit contexts (cell references, formulas, and layout) to convey meaning. Natural language processing and machine learning methods struggle with this rich but irregular format, which calls for specialized resources and models to advance spreadsheet understanding.

Despite the widespread use of spreadsheets, large-scale corpora for spreadsheet AI research remain scarce. Existing datasets, such as EUSES (~4,000 files) [1], Enron (~15,000 files) [2], and Fuse (~249,000 files) [3], are limited by small scale, narrow domains, or a lack of formula-rich content. Many contain static data or simple tables, and accessing their raw data can be challenging. Sheetpedia addresses this gap by curating a comprehensive collection of spreadsheets from multiple sources, yielding a corpus of ~290,000 unique worksheets – to our knowledge, the largest publicly available

---

*Corresponding author.

39th Conference on Neural Information Processing Systems (NeurIPS 2025) Track on Datasets and Benchmarks.

spreadsheet corpus to date. Our dataset encompasses varied domains (financial models, inventory lists, academic data, etc.), and through careful preprocessing we ensure a high quality of content (e.g. language filtering yields 78% English sheets, and deduplication cuts 48% of near-duplicates). This diverse, large-scale corpus provides a rich foundation for training and evaluating AI models on real spreadsheet structures and formulas.

Another key contribution of Sheetpedia is demonstrating how such data can be leveraged to improve model performance on novel spreadsheet-related tasks. While general table understanding has seen progress (e.g. representation learning on Wikipedia tables in TURL [4], or semantic parsing to SQL queries in Spider [5]), spreadsheets pose unique challenges that require new problem formulations. We focus on two tasks that highlight natural language interaction with spreadsheets: (1) Natural Language to Semantic Range (NL2SR) – mapping a user's plain-language request to the correct cell range or region in a spreadsheet; and (2) Natural Language to Formula (NL2Formula) – generating a valid Excel formula that fulfills a given natural language description. These tasks simulate practical scenarios, such as a user asking "What is the total sales for Q1?" (NL2SR would identify the relevant cells to sum, and NL2Formula would produce the SUM formula over that range). Both tasks are challenging because they require understanding the semantics of the spreadsheet content (schema, headings, data distributions) as well as the natural language query. They go beyond standard table QA: NL2Formula in particular is a code-generation problem (synthesizing formula code), which has only recently begun to be studied in NLP.

A major hurdle in tackling NL2SR and NL2Formula is the lack of large annotated training datasets. Recent work by [6] introduced an NL2Formula benchmark of 70K NL-formula pairs by converting existing text-to-SQL examples into spreadsheet context. This conversion approach, while clever, is limited by the coverage of SQL patterns and does not address the NL2SR task. In our work, we propose a complementary data generation strategy leveraging LLMs themselves to bootstrap training data. We employ a rejection sampling framework, utilizing a specialized judge model [7] to curate high-quality synthetic training data at scale. Specifically, we prompt a LLM to generate candidate formulas or ranges for novel natural language queries. These candidates are then evaluated by the judge model, which filters out incorrect or inconsistent outputs, ensuring a robust and reliable dataset. This approach is inspired by successes in code generation, where sampling multiple outputs and selecting correct ones can dramatically improve accuracy. For example, OpenAI's Codex model [8] (a GPT-3 [9] variant for code) solved 70% of programming tasks with 100 samples vs. only 28% with one try. By applying iterative self-refinement via rejection sampling, we build sizeable training sets for NL2SR and NL2Formula without extensive manual labeling.

We fine-tune state-of-the-art LLMs on Sheetpedia for these tasks and find that spreadsheet-specific training yields substantial gains. Our NL2SR model accurately identifies the correct cell ranges for user queries in 97.5% of test cases, and our NL2Formula model achieves 71.7% exact formula generation accuracy. These results underscore the value of domain-specific data: general-purpose models like Codex or GPT-3, while powerful, benefit greatly from fine-tuning on targeted spreadsheet data and tasks. Indeed, specialized models for spreadsheet formulas (e.g. the 60M-parameter FLAME model [10] ) have been shown to outperform much larger general code models on formula prediction tasks, reinforcing our findings.

In summary, our work makes the following contributions: (1) We introduce Sheetpedia, a large-scale, diverse, and formula-rich spreadsheet corpus comprising over 290,000 worksheets, addressing the limitations of prior datasets in scale, quality, and semantic richness; (2) We define and release benchmarks for two novel spreadsheet understanding tasks—Natural Language to Semantic Range (NL2SR) and Natural Language to Formula (NL2Formula)—grounded in realistic spreadsheet usage; and (3) We demonstrate that fine-tuning LLMs on Sheetpedia, combined with a rejection sampling-based data generation framework, significantly improves task performance, achieving up to 97.5% accuracy on NL2SR and 71.7% on NL2Formula.

## 2   Related Work

**Spreadsheet and Table Datasets.** Early spreadsheet datasets, such as the EUSES Spreadsheet Corpus, provided a collection of approximately 4,000 real-world spreadsheets sourced from the web. However, EUSES is limited in scale, and many files lack formulas, reducing their utility for training formula-centric models. The Enron spreadsheet dataset, derived from the Enron email

release, offers around 15,000 spreadsheets with richer formula content (60% include formulas) but is confined to a single corporate domain.More recently, the Fuse dataset mined 249,000 spreadsheets from the Common Crawl, but only 7% contain formulas, and many files are incomplete or require reconstruction. The DECO dataset [11] provides 1,165 annotated spreadsheets from the Enron corpus, with cell-level layout annotations for table recognition tasks. The DeExcelerator framework[12], evaluated on datasets like DeEx, CIUS, and SAUS, extracts relational data from semi-structured spreadsheets and HTML tables. Beyond spreadsheets, large table corpora, such as Wikipedia tables used in TURL (1.6M tables) and Google's TaPas [13] (millions of tables), support tasks like table question-answering but lack the dynamic formulas and user-generated context inherent to spreadsheets. Sheetpedia addresses these limitations by combining web-crawled data (inspired by Fuse), enterprise data (like Enron), and a scrape of 147,000 user-contributed spreadsheets from ExcelForum.

**Spreadsheet Intelligence and Formula Synthesis.** Research on intelligent spreadsheet assistance has focused on formula prediction and synthesis. Microsoft's FlashFill [14] infers string transformation formulas from user examples, a form of programming by example. More advanced approaches, such as SpreadsheetCoder [15], use neural networks to predict formulas based on cell context, treating the task as code completion. In the NLP community, semantic parsing techniques inspired by Text-to-SQL datasets like WikiSQL [16] and Spider have been adapted to spreadsheets. The NL2Formula task, for instance, involves generating Excel formulas from English queries, with a dataset of 70,000 query-formula pairs derived from Text-to-SQL problems. Complementary to this, we introduce NL2SemanticRange (NL2SR), a novel task for predicting cell range addresses from natural language queries, extending prior work on natural language information retrieval in spreadsheets, such as NLP-SIR [17]. Together, NL2Formula and NL2SR formalize core aspects of spreadsheet interaction—data selection and formula generation—enabling comprehensive intelligent assistance.

**Large Language Models for Tables and Spreadsheets.** The rise of LLMs has spurred interest in their application to structured data. Models like TURL and TaPas pre-train transformers on millions of relational tables to enhance tasks such as column type annotation and table question-answering. However, these models focus on static tables and lack spreadsheet-specific grounding, such as understanding cell ranges or formulas. To bridge this gap, FORTAP [18] incorporates formulas during pre-training to make the model aware of numerical reasoning over tables. In contrast, Codex, a GPT-3-based code model, can generate Excel formulas from natural language but struggles with spreadsheet layouts without examples. Specialized models like FLAME , a 60M-parameter transformer trained on Excel formulas, outperform larger general-purpose models on formula repair and completion by leveraging domain-specific data. Our work aligns with this trend, demonstrating that fine-tuning LLMs on Sheetpedia's rich spreadsheet corpus significantly improves performance on spreadsheet tasks compared to generic models. Additionally, our rejection sampling methodology for data generation draws inspiration from program synthesis techniques, where multiple candidate outputs are tested for correctness.

# 3 Sheetpedia

To build a high-quality spreadsheet corpus for NLP tasks, we integrate diverse sources, including public datasets and user-uploaded content from professional forums. This section outlines the workflow, which involves sourcing data from public datasets and ExcelForum, followed by a preprocessing pipeline that standardizes formats, cleans and filters content, and deduplicates worksheets to create a robust corpus of 290,509 unique worksheets. The following subsections introduce these in detail.

## 3.1 Data Sources

The corpus draws from two primary sources: existing public datasets and ExcelForum (`https://www.excelforum.com/`), a leading platform for technical spreadsheet discussions. These sources span applications like financial modeling and data analysis, ensuring a broad representation of spreadsheet use cases.

**Public Datasets.** It includes two public datasets. The Fuse dataset contains 161,323 workbooks (182,784 worksheets[1]) sourced from the internet, offering diverse content. The Enron dataset provides 15,927 workbooks (62,612 worksheets[1]) from corporate emails, reflecting enterprise contexts.

**ExcelForum.** A custom crawler collected 147,738 workbooks (320,489 worksheets) from ExcelForum, a platform with over 1 million threads and 5 million posts as of May 2024. Targeting the "Excel General" and "Excel Formulas & Functions" subforums, the crawler adhered to `robots.txt`, avoiding restricted directories (e.g., `search.php`) and bot-related files (e.g., Bytespider).

The raw corpus totals 324,988 workbooks and 566,018 worksheets, serving as a substantial resource for NLP and document intelligence research.

## 3.2 Preprocessing Pipeline

A comprehensive preprocessing pipeline standardizes data, enhances quality, and eliminates redundancies through format conversion, cleaning, language filtering, and deduplication.

**Format Standardization.** All `.xls` files are converted to the modern `.xlsx` format using `pyexcel`, as the newer XML-based format provides better data integrity and compatibility with contemporary tools. The `openpyxl` library then extracts cell contents, formulas, and metadata, serializing them into JSON with fields for filename, sheetname, UsedRange, Cells, and others.

**Data Cleaning.** First, language filtering uses `lingua` to identify dominant languages. For each worksheet, we concatenate non-empty, non-numeric, and non-formula cells (e.g., excluding cells starting with =) row-wise to form detection contexts. Worksheets with fewer than 20 valid cells are excluded as linguistically unverifiable. Language detection applies a confidence threshold of 0.8, with mixed-language detection and a lower threshold (0.5) retry for ambiguous cases to reduce "unknown" classifications. Across 295,672 valid worksheets, English dominates (77.40%, 228,843 worksheets), followed by "unknown" (13.34%) and minor languages like Latin (1.76%).

Second, formulas are filtered to retain only those that are syntactically valid and functionally complex. We exclude formulas with cross-sheet references (e.g., Sheet2!A1) and those consisting solely of a single text-manipulation function (e.g., CONCAT, TEXTJOIN, LEFT) without range references. Retained formulas must contain at least one valid Excel function (e.g., SUM, VLOOKUP, IF) from a predefined set of standard official functions and reference non-empty cells within the same worksheet. Syntactic correctness is verified by tokenizing formulas and validating cell ranges (e.g., A1, A1:B10).

**Spreadsheet Deduplication.** To eliminate redundant worksheets in the corpus, we employ a deduplication pipeline based on MinHash [19] and Locality-Sensitive Hashing (LSH) [20], leveraging Jaccard similarity as the core metric. Spreadsheets are naturally suited for set-based similarity analysis: each worksheet can be represented as a set of non-empty, non-numeric cell values (excluding formulas), capturing its textual content. Jaccard similarity, defined as the size of the intersection divided by the union of two sets, effectively measures content overlap between worksheets, making it ideal for identifying near-duplicates in this context.

Initially, we extracted non-empty, non-numeric cells from each worksheet, excluding worksheets with fewer than 20 valid cells to guarantee sufficient content for reliable comparisons. For the remaining worksheets, we computed MinHash signatures (length = 1000) to efficiently approximate Jaccard similarities. These signatures were clustered using LSH with a Jaccard similarity threshold of 0.8, effectively grouping worksheets exhibiting significant content overlap. The chosen LSH configuration employed parameters of r=100 and b=10, carefully balancing precision and recall. Specifically, the high value of r ensured precision by demanding strong within-band similarity, thus minimizing false positives (dissimilar worksheets incorrectly clustered). Concurrently, a moderate value of b maintained acceptable recall, effectively capturing most near-duplicates without incurring excessive computational costs. The selected configuration of r=100, b=10 significantly reduced the corpus size from 566,018 to 290,509 worksheets (a 48.7% reduction).

---

[1]Numbers reflect workbooks/worksheets that were successfully parsed and accessible; some files were excluded due to corruption or format obsolescence.

[2]`https://github.com/pyexcel/pyexcel`

[3]`https://openpyxl.readthedocs.io/en/stable/`

[4]`https://github.com/pemistahl/lingua-py`.

An alternative configuration (r=10, b=100) yielded excessive clustering and missed genuine duplicates. Our selected configuration (r=100, b=10), validated via manual inspection, ensured high precision and sufficient recall. The Union-Find algorithm finalized clustering in approximately 9 minutes, producing 290,509 unique worksheets: Enron (62,612 → 28,032), Fuse (182,784 → 105,097), and ExcelForum (320,489 → 157,380).

## 4  Dataset Statistics

This section analyzes the final deduplicated spreadsheet corpus of 290,509 worksheets to characterize its scale, diversity, and suitability for NLP tasks. The analysis examines formula pattern distributions, corpus characteristics at workbook and worksheet levels, and language distribution. Through statistical summaries, visualizations, and language detection, we highlight the corpus's prevalent formula patterns, skewed distributions and English dominance, providing insights into its structure and utility.

### 4.1  Formula Pattern Distribution

We analyzed the distribution of individual functions and their co-occurrence patterns in formulas. Non-arithmetic formula patterns are predominantly led by IF, MATCH, IFERROR, and AND (Figure1a). The co-occurrence network (Figure1a) reveals that functions such as IF, SUM, VLOOKUP, and COUNTIF occupy important positions with larger nodes and numerous connections, indicating their high usage frequency and frequent combination with other functions. This aligns with common knowledge, as these functions are foundational in Excel, serving essential purposes like conditional logic (IF), aggregation (SUM), and data lookup (VLOOKUP). Notably, IFERROR exhibits strong connections with IF and VLOOKUP, reflecting its common use in handling potential errors from these functions, such as wrapping VLOOKUP to avoid error values—a practical pattern observed in real-world applications.

The word cloud (Figure 1b) further highlights the prominence of IF, SUM, and VLOOKUP, consistent with their dominance in the co-occurrence network. Additionally, the presence of terms like ERROR and ISERROR in the word cloud underscores error handling as a critical aspect of formula design, corroborating the frequent use of IFERROR observed in the network.

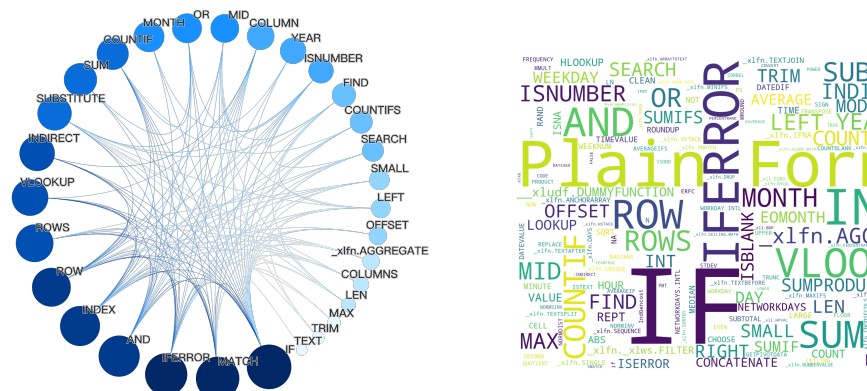

(a) Co-occurrence network of the top 30 functions.  (b) Word cloud visualizing formula structure patterns.

Figure 1: Visualization of formula pattern distribution

### 4.2  Spreadsheet Corpus Characteristics

Table 1 summarizes the spreadsheet corpus at workbook and worksheet levels, covering cell, row, column, and worksheet counts. The data reveals a highly skewed distribution, with most spreadsheets being small but a few exhibiting extreme sizes. At the workbook level, cell counts range from 20 to 60,614,911 (mean: 21,290.72, median: 270), indicating that most workbooks are compact, with 75% having 1,218 or fewer cells (Q3). Row (median: 67) and column (median: 10) counts show similar skewness, and worksheet counts (mean: 1.37, median: 1) confirm that single-sheet workbooks

dominate. At the worksheet level, cell counts (mean: 15,487.37, median: 300) and row/column counts (medians: 48 and 10) reflect compact worksheets, with 75% having 1,161 or fewer cells.

Table 1: Statistical summary of the spreadsheet corpus at workbook and worksheet levels.

| Level | Metric | Min | Max | Mean | Median | Q1 | Q3 | Mode |
|---|---|---|---|---|---|---|---|---|
| Workbook | Cell Count | 20 | 60,614,911 | 21,290.72 | 270 | 201 | 1,218 | 201 |
| | Row Count | 1 | 3,145,590 | 931.58 | 67 | 23 | 87 | 67 |
| | Column Count | 1 | 98,304 | 49.07 | 10 | 3 | 20 | 3 |
| | Worksheet Count | 1 | 123 | 1.37 | 1 | 1 | 1 | 1 |
| Worksheet | Cell Count | 20 | 60,614,872 | 15,487.37 | 300 | 184 | 1,161 | 201 |
| | Row Count | 1 | 1,048,576 | 677.65 | 48 | 21 | 75 | 67 |
| | Column Count | 1 | 16,384 | 35.69 | 10 | 4 | 17 | 3 |

## 4.3 Language Distribution

The language distribution of 247,909 worksheets, identified using the `lingua` library after filtering, is highly skewed. English dominates, comprising 78.85% of the dataset (195,479 worksheets), followed by an "Unknown" category at 11.99% (29,726 worksheets). The significant Unknown portion likely reflects worksheets with minimal text, mixed languages, or formats challenging for automated detection. Among the 70 other identified languages, Latin (1.78%, 4,423 worksheets) and Yoruba (1.73%, 4,281 worksheets) are the most prevalent minor languages, followed by 68 others (e.g., German: 0.44%, Spanish: 0.43%) collectively contributing 9.16%. This distribution highlights the predominance of English-language materials, with moderate linguistic diversity across a long tail of minor languages.

## 5 Spreadsheet Corpus for Downstream Tasks

This section presents a spreadsheet corpus designed to enhance LLM performance on spreadsheet-related tasks through fine-tuning. We focus on two tasks: Natural Language to Semantic Range (NL2SR) and Natural Language to Formula (NL2Formula). To address the challenge of acquiring high-quality training and test data, we employ distinct data generation strategies: rejection sampling for the training set and a combination of LLM generation with human review for the test set. Fine-tuned models demonstrate significant improvements, achieving up to 97.50% accuracy for NL2SR and 71.67% for NL2Formula, validated through comprehensive experiments.

### 5.1 Task Definitions

This work focuses on two spreadsheet-specific tasks: Natural Language to Semantic Range (NL2SR) and Natural Language to Formula (NL2Formula). These tasks are designed to evaluate a model's ability to interpret and manipulate spreadsheets, capturing their structural and semantic complexity. NL2SR involves mapping natural language queries to precise cell ranges in a spreadsheet, evaluated by the accuracy of the identified range. NL2Formula requires generating syntactically correct and semantically accurate formulas from natural language queries, assessed by the correctness of the formula's output in context.

These tasks are fundamental because spreadsheets combine structured data (e.g., cell grids, references) with semantic intent (e.g., calculations, aggregations). NL2SR tests a model's understanding of spatial relationships and data organization within the table structure, critical for tasks like data selection or filtering. NL2Formula evaluates the ability to translate semantic intent into executable formulas, requiring both syntactic precision and contextual reasoning. High-quality training and test data are essential to address these challenges, as they enable models to learn the intricate interplay of structure and semantics, overcoming the primary bottleneck in spreadsheet intelligence. Examples of these tasks are illustrated in Figure 2.

### 5.2 Dataset Construction

We construct a spreadsheet corpus comprising training, validation, and test sets, using tailored strategies to ensure data quality for NL2SR and NL2Formula tasks.

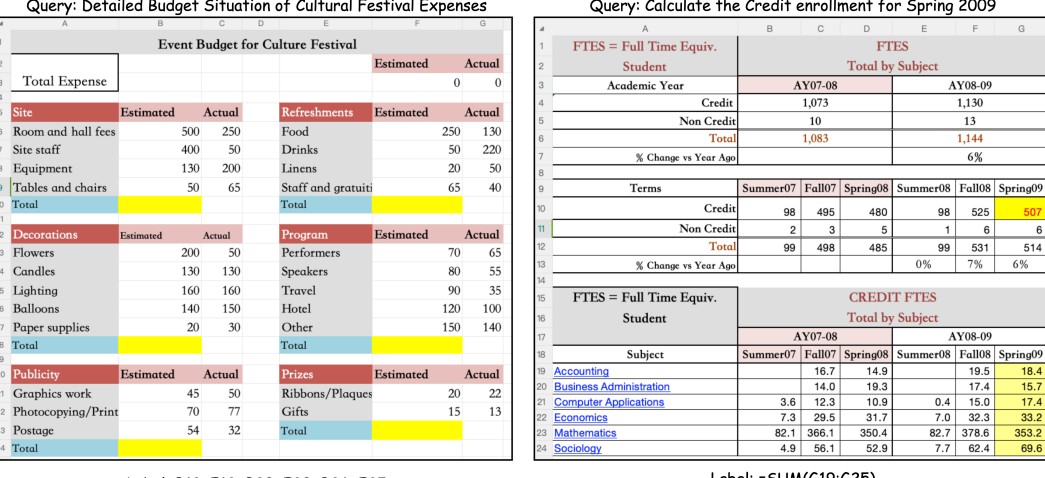

Figure 2: Examples of NL2SR (left) and NL2Formula (right) tasks. The left side illustrates mapping a natural language query to a semantic cell range, while the right side shows generating a formula from a natural language query.

**Training Set Generation** The training set is constructed through an iterative rejection sampling framework (Algorithm 1). For each spreadsheet target (formula or range), we generate a context prompt and sample $k = 5$ candidate queries using Gemini-Flash-2.0 [21](generation temperature $\tau_g = 0.7$), balancing diversity and coherence. Each candidate is evaluated by Claude-3.7-Sonnet [22] following the LLM-as-a-Judge protocol, scoring four dimensions: clarity (30%), accuracy (30%), conciseness (20%), and completeness (20%). Candidates achieving a composite quality score of at least $\gamma = 0.7$ are retained for dataset augmentation.

**Test Set Generation** The test set, consisting of 240 samples, is generated using Claude-3.7-Sonnet for both query generation and scoring, following a process similar to the training set but with an additional human review step. Queries are generated with the same parameters ($k = 5$, $\tau_g = 0.7$) and scored using the same four-dimensional criteria. Human experts review and refine the queries to ensure correctness and relevance, enhancing the test set's reliability for benchmarking.

---

**Algorithm 1** Iterative Query Generation Algorithm

---

**Require:** Original spreadsheet dataset $\mathcal{D}$, generation model $M_g$, scoring model $M_s$
**Ensure:** Augmented dataset $\mathcal{D}_{aug}$
1: **for** each spreadsheet $T \in \mathcal{D}$ **do**
2:     **for** each target $o \in T.targets$ **do**
3:         */* Target: formula (Formula) or range (Range) */*
4:         Construct context prompt $p \leftarrow$ generate_prompt$(T, o)$
5:         Generate candidate queries $\{q_i\}_{i=1}^{k} \leftarrow M_g(p)$
6:         Compute quality scores $s_i \leftarrow M_s(q_i, o, T)$
7:         Filter high-quality samples $\mathcal{Q} \leftarrow \{q_i \mid s_i \geq \gamma\}$
8:         Add training pairs $\mathcal{D}_{aug} \leftarrow \mathcal{D}_{aug} \cup \{(q, o, T) \mid q \in \mathcal{Q}\}$
9:     **end for**
10: **end for**

---

**Dataset Statistics** The training and validation sets are split in a 9:1 ratio, with statistics summarized in Table 2. The test set is curated separately to ensure diversity and quality.

Table 2: Dataset Statistics for NL2Formula and NL2SemanticRange Tasks (Token Lengths)

| Task | Input Length | | | | Output Length | | | |
|------|------|-----|-----|-----|------|-----|-----|-----|
| | **Mean** | **Min** | **Max** | **Med** | **Mean** | **Min** | **Max** | **Med** |
| **NL2Formula** | | | | | | | | |
| Train ($n = 1,957$) | 4,400 | 474 | 16,382 | 2,904 | 28 | 13 | 335 | 21 |
| Valid ($n = 210$) | 4,405 | 502 | 16,194 | 3,192 | 25 | 13 | 90 | 19 |
| **NL2SemanticRange** | | | | | | | | |
| Train ($n = 2,204$) | 5,076 | 1,220 | 16,364 | 3,758 | 16 | 15 | 19 | 17 |
| Valid ($n = 239$) | 5,103 | 1,277 | 16,376 | 3,782 | 16 | 15 | 19 | 17 |

## 5.3 Experimental Setup

This subsection outlines the fine-tuning strategies, baseline models, and evaluation metrics used to assess model performance on the NL2SR and NL2Formula tasks.

**Fine-Tuning Strategies**  Fine-tuning is performed on an NVIDIA 8xA800 GPU setup using two approaches. The first, LoRA[23], employs rank 16, targeting all parameters, with the AdamW optimizer (learning rate $2 \times 10^{-5}$, cosine schedule, 0.1 warmup ratio), batch size 1 per device, 8 gradient accumulation steps, 5 epochs, and bf16 precision. The second, full-parameter fine-tuning, uses the same hyperparameters with DeepSpeed optimizations for efficiency. We evaluate single-task fine-tuning (NL2SR or NL2Formula) and mixed-data fine-tuning (both tasks) to analyze multi-task performance trade-offs.

**Baseline Models**  The evaluation includes five baseline models selected for their diversity in scale, architecture, and accessibility. LLaMA-3.1-8B-Instruct [24] and Qwen2.5-7B-Instruct [25] are efficient open-source models, valued for academic and industrial applications, with Qwen2.5-7B offering strong multilingual capabilities. LLaMA-3.3-70B and Qwen2.5-72B, larger open-source models, provide insights into scaling effects. GPT-4o[26], a powerful closed-source model, serves as a high-performance benchmark. This selection balances computational feasibility, open-source availability, and cutting-edge performance, enabling a comprehensive comparison across model sizes and training paradigms.

**Evaluation Metrics**  Performance is measured using accuracy, defined as: (1) for NL2SR, the proportion of correct range mappings; (2) for NL2Formula, the proportion of formulas that exactly match the ground truth (exact match required). Both tasks are evaluated on a 240-sample test set.

## 5.4 Experimental Results

**Few-Shot Performance**  Table 3 compares the accuracy of baselines in zero-shot, one-shot, and three-shot settings across NL2SR and NL2Formula tasks, based on 120 test samples per task. Here, Qwen2.5-72B achieves the highest NL2SR accuracy (73.33–75.83%), surpassing GPT-4o

Table 3: Accuracy Comparison of Models in Few-Shot Settings (%)

| Model | NL2SR | | | NL2Formula | | |
|-------|--------|--------|--------|--------|--------|--------|
| | **0-shot** | **1-shot** | **3-shot** | **0-shot** | **1-shot** | **3-shot** |
| GPT-4o | 66.67 | 69.17 | 65.83 | **63.33** | **64.17** | **63.33** |
| LLaMA-3.1-8B | 41.67 | 28.33 | 43.33 | 27.50 | 25.00 | 26.67 |
| Qwen2.5-7B | 45.83 | 51.67 | 54.17 | 30.83 | 33.33 | 40.00 |
| LLaMA-3.3-70B | 66.67 | 67.50 | 70.00 | 47.50 | 59.17 | 57.50 |
| Qwen2.5-72B | **75.83** | **74.17** | **73.33** | 60.00 | 59.17 | 61.67 |

(65.83–69.17%) across all few-shot settings. For NL2Formula, GPT-4o leads (63.33–64.17%), with Qwen2.5-72B close behind (59.17–61.67%). Smaller models, LLaMA-3.1-8B and Qwen2.5-7B, show lower performance (25.00–54.17%), indicating limitations in zero-shot and few-shot generalization.

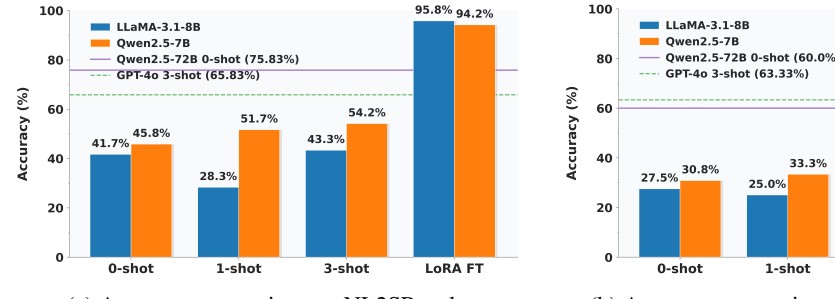

(a) Accuracy comparison on NL2SR task.      (b) Accuracy comparison on NL2Formula task.

Figure 3: Performance of LoRA fine-tuning versus few-shot learning. For both NL2SR and NL2Formula tasks, LoRA fine-tuning provides a substantial accuracy boost, outperforming all few-shot settings and surpassing strong baselines from larger models.

**Fine-Tuning Results** Figures 3a and 3b compare LLaMA-3.1-8B and Qwen2.5-7B across zero-shot, one-shot, three-shot, and LoRA fine-tuning settings. For NL2SR, Qwen2.5-7B outperforms LLaMA-3.1-8B in few-shot settings, with both reaching near 95% accuracy after LoRA fine-tuning. For NL2Formula, Qwen2.5-7B maintains a slight edge, both surpass the performance of GPT-4o.

To further explore LLaMA-3.1-8B's fine-tuning strategies, Figure 4 visualizes the test accuracy of LoRA and Full-Param approaches across NL2SR, NL2Formula, and Mixed data types for both tasks, with darker colors indicating higher performance. LoRA fine-tuning on NL2SR data achieves the highest NL2SR accuracy (97.50%) but significantly degrades NL2Formula performance (5.00%), as shown in Figure 4. Conversely, Full-Param fine-tuning on NL2Formula data yields the best NL2Formula accuracy (71.67%) while maintaining competitive NL2SR performance (37.50%). Mixed-data strategies, particularly Full-Param, balance both tasks effectively (96.67% NL2SR, 70.00% NL2Formula), highlighting the dataset's quality and the trade-offs between LoRA's efficiency and Full-Param's multi-task robustness.

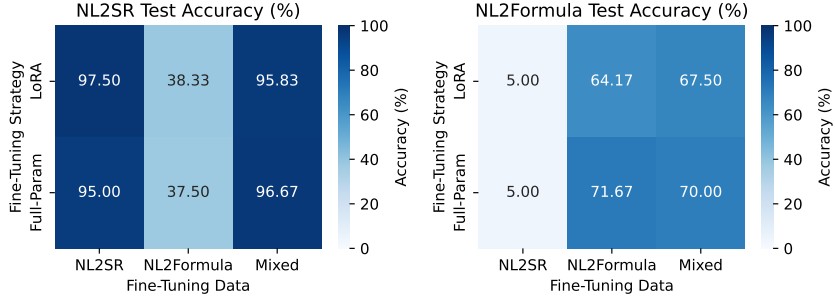

Figure 4: Heatmaps comparing fine-tuning strategies (LoRA and Full-Param) on LLaMA-3.1-8B across NL2SR, NL2Formula, and Mixed data types for NL2SR and NL2Formula tasks.

# 6 Conclusion

We introduced Sheetpedia, a large-scale, diverse, and formula-rich spreadsheet corpus comprising over 290,000 unique worksheets, significantly surpassing existing datasets in scale, diversity, and semantic richness. Sheetpedia serves as a valuable resource for spreadsheet intelligence, enabling the development and benchmarking of advanced NLP models through two newly introduced tasks: Natural Language to Semantic Range (NL2SR) and Natural Language to Formula (NL2Formula). The rejection sampling-based fine-tuning strategy demonstrated strong performance gains, achieving 97.5% accuracy on NL2SR and 71.7% on NL2Formula. Sheetpedia, along with its benchmarks and methods, sets a new foundation for future research in spreadsheet understanding, fostering improvements in both theoretical modeling and practical application.

# 7 Limitations

Despite these advancements, our work also faces certain limitations. The corpus predominantly contains English-language spreadsheets, potentially limiting applicability to multilingual contexts. Additionally, our deduplication and formula filtering methods might omit smaller spreadsheets or complex cross-sheet formulas, which may impact dataset diversity and representativeness. Furthermore, while our benchmarks and tasks are comprehensive, they may not fully capture all intricacies and challenges encountered in real-world spreadsheet usage. Future research could extend Sheetpedia by incorporating multilingual data, refining deduplication and formula inclusion methods, and broadening task coverage to enhance the dataset's generalizability and practical relevance.

## Acknowledgments and Disclosure of Funding

The authors wish to thank the anonymous reviewers for their constructive feedback. This work was supported by the Beijing Natural Science Foundation (No. L253020) and the National Natural Science Foundation of China (No. 62036001). This research was also supported by the National Research Foundation, Singapore under its National Large Language Models Funding Initiative (AISG Award No. AISG-NMLP-2024-002), and by the Ministry of Education, Singapore, under its AcRF Tier 2 Funding (Proposal ID: T2EP20123-0052). Any opinions, findings, conclusions, or recommendations expressed in this material are those of the author(s) and do not reflect the views of the National Research Foundation or the Ministry of Education, Singapore.

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

# 8 Technical Appendices and Supplementary Material

This appendix provides supplementary materials to support the main findings and methodologies presented in the Sheetpedia corpus study. It includes detailed visualizations, statistical tables, and prompt templates used for data generation and evaluation. These materials offer deeper insights into the corpus's characteristics, formula distributions, language diversity, and the construction of the NL2SR and NL2Formula tasks. Each figure and table is accompanied by a brief description to guide readers through the content, ensuring accessibility and clarity for researchers aiming to replicate or extend our work.

**Data Sanitization and PII Masking**   To ensure data privacy and ethical compliance, we implemented a comprehensive pipeline to identify and mask Personally Identifiable Information (PII) within the corpus. The process was executed using a Python script leveraging multi-core processing via the `concurrent.futures` library to efficiently handle the large volume of XLSX files. For each spreadsheet, the script systematically iterated through every cell and applied a series of masking rules.

The core of this process involved a multi-faceted approach to PII detection and replacement:

- **Person Names:** We utilized the `spaCy` natural language processing library (specifically, the `en_core_web_sm` model) for Named Entity Recognition (NER). Detected person names were replaced with a consistent, file-specific placeholder. For instance, the first unique name found in a file would be replaced with `[Person_1]`, the second with `[Person_2]`, and so on. This method ensures that all occurrences of the same name within a single document are mapped to the same placeholder, preserving contextual integrity while guaranteeing anonymity.

- **Email Addresses:** A regular expression was used to identify email addresses. To balance privacy with the potential utility of domain information, we masked the local-part (username) of the email with a generic `[USER]` token while preserving the domain (e.g., `example@edu.com` becomes `[USER]@edu.com`).

- **Phone Numbers:** Phone numbers were also detected using regular expressions. The script masked the initial digits with a `[PHONE_PREFIX]` token but retained the final four digits. This approach removes the identifiable portion of the number while leaving a non-identifiable suffix that could potentially be used for record linkage without compromising privacy.

**Overview of Language Distribution**   Figure 5 visualizes the language distribution of the deduplicated Sheetpedia corpus, highlighting the predominance of English and the presence of minor languages. This chart complements the language statistics in Section 4.3, providing a clear visual representation of linguistic diversity.

**Formula Pattern Insights**   Figure 6 illustrates the distribution of the top 20 Excel formula patterns in the deduplicated corpus, using a logarithmic scale to emphasize the prevalence of common functions like `IF` and `SUM`. This visualization supports the analysis in Section 4.1, revealing key patterns in formula usage.

**Language Distribution Details**   The table below presents the detailed language distribution of the deduplicated Sheetpedia corpus, listing the top 15 languages by count and percentage. This complements Figure 5 and Section 4.3, offering precise statistics for researchers studying linguistic diversity in spreadsheet data.

**Formula Usage Before Deduplication**   This table lists the top 30 Excel formula patterns in the raw corpus before deduplication, showcasing the frequency of key functions like `IF` and `SUM`. It provides a baseline for understanding formula prevalence, as discussed in Section 4.1.

**Formula Usage After Deduplication**   This table details the top 30 Excel formula patterns in the deduplicated corpus, reflecting changes in frequency after removing redundant worksheets. It supports the analysis in Section 4.1 and highlights the impact of deduplication on formula distribution.

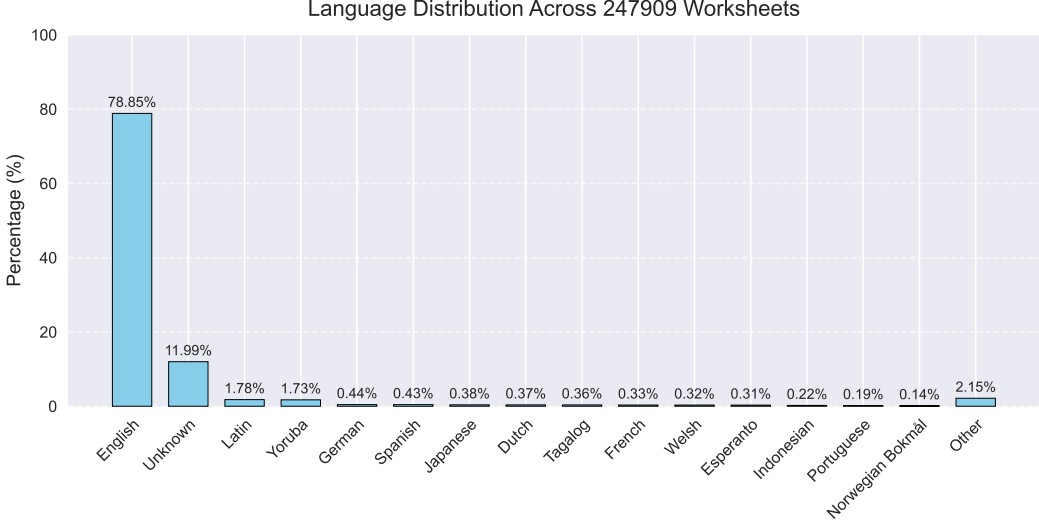

Figure 5: Language distribution of deduplicated corpus (percentage)

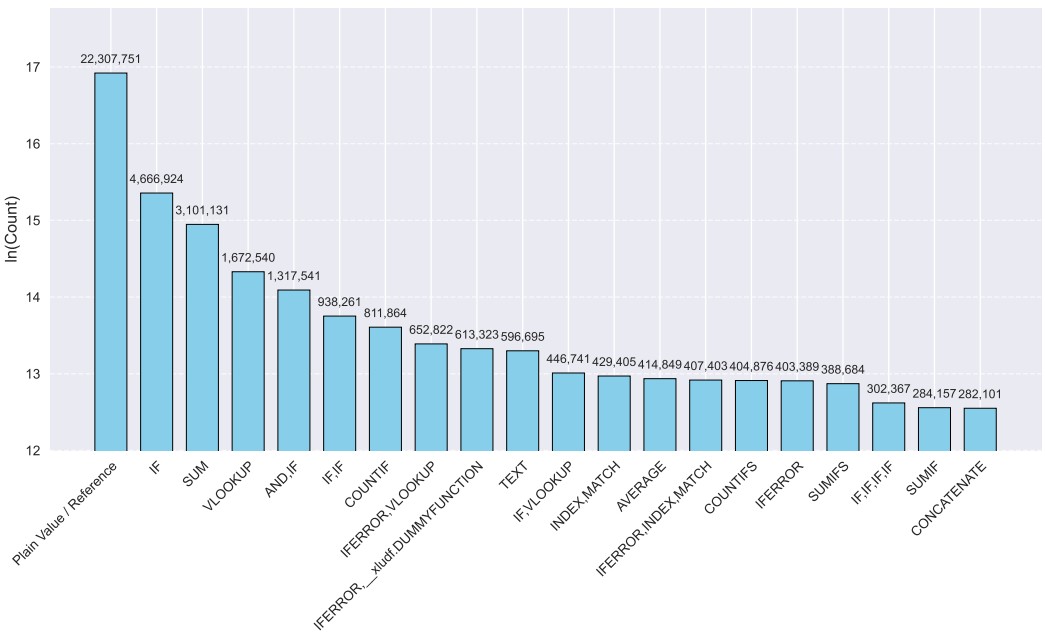

Figure 6: Distribution of formula patterns in deduplicated corpus

Table 4: Language distribution in the corpus (Top 15)

| Language Code | Count | Percentage (%) |
|---|---|---|
| en | 195,479 | 78.85 |
| unknown | 29,726 | 11.99 |
| la | 4,423 | 1.78 |
| yo | 4,281 | 1.73 |
| de | 1,093 | 0.44 |
| es | 1,061 | 0.43 |
| ja | 939 | 0.38 |
| nl | 915 | 0.37 |
| tl | 901 | 0.36 |
| fr | 818 | 0.33 |
| cy | 793 | 0.32 |
| eo | 768 | 0.31 |
| id | 554 | 0.22 |
| pt | 462 | 0.19 |
| nb | 357 | 0.14 |

**Distribution of Merged Cells**   To understand the structural complexity of the spreadsheets within the corpus, we analyzed the prevalence of merged cells—a common feature used for formatting and presentation. Our analysis of all 290,509 successfully processed worksheets revealed that 56,494 of them contained at least one merged cell. This corresponds to approximately 19.45% of the total worksheets. The significant presence of merged cells highlights a key challenge in automated spreadsheet processing, as such structures can complicate data extraction and table recognition. This finding underscores the necessity for robust parsing techniques that can accurately interpret visually formatted layouts, a critical consideration for developing effective spreadsheet intelligence models.

**Prompt for NL2SR Query Generation**   The prompt shown in Figure 7 outlines the instructions provided to the language model for generating natural language queries for the NL2SR task. It specifies the input format, guidelines for query precision, and output structure, as described in Section 5.2.

**Prompt for NL2Formula Query Generation**   The prompt in Figure 8 details the instructions for generating natural language queries for the NL2Formula task, specifying the input format and requirements for capturing formula semantics. This supports the dataset construction process outlined in Section 5.2.

**Prompt for NL2SR Query Scoring**   The prompt in Figure 9 specifies the criteria and scoring rubric for evaluating the quality of NL2SR queries, as used in the dataset construction pipeline (Section 5.2). It ensures queries meet standards for clarity, accuracy, conciseness, and completeness.

**Prompt for NL2Formula Query Scoring**   The prompt in Figure 10 defines the evaluation criteria for NL2Formula queries, ensuring they accurately reflect formula logic, as described in Section 5.2.

**Prompt for NL2SR Testing**   The prompt in Figure 11 provides the instructions for testing models on the NL2SR task, specifying how to map user queries to cell ranges, as used in Section 5.3.

**Prompt for NL2Formula Testing**   The prompt in Figure 12 outlines the instructions for testing models on the NL2Formula task, detailing how to generate formulas from user queries, as described in Section 5.3.

**Chain-of-Thought (CoT) Prompting Analysis**   To assess the applicability of Chain-of-Thought (CoT) prompting for spreadsheet-related tasks, we conducted an exploratory analysis on NL2SR and NL2Formula using baseline LLMs in a zero-shot setting. CoT was induced by appending the phrase "Please think step by step." to the standard prompts. The results, summarized in Table 7, show that CoT does not consistently improve performance and, in many cases, degrades accuracy compared

Table 5: Top 30 Most Used Excel Formulas: Before Deduplication

| Formula | Count |
|---|---|
| Plain Formula | 30,798,987 |
| IF | 8,121,168 |
| SUM | 5,723,627 |
| AND,IF | 3,155,740 |
| VLOOKUP | 2,547,198 |
| IF,IF | 1,562,982 |
| IFERROR,VLOOKUP | 1,163,168 |
| COUNTIF | 1,162,861 |
| TEXT | 939,481 |
| AVERAGE | 824,825 |
| INDEX,MATCH | 814,055 |
| IFERROR,INDEX,MATCH | 784,835 |
| IFERROR,_xludf.DUMMYFUNCTION | 781,777 |
| IFERROR | 656,478 |
| IF,VLOOKUP | 639,355 |
| IF,IF,IF,IF | 567,283 |
| SUMIF | 540,015 |
| CHAR,CHAR,CLEAN,SUBSTITUTE,SUBSTITUTE,TRIM | 539,178 |
| SUMIFS | 534,398 |
| COUNTIFS | 519,826 |
| CONCATENATE | 510,803 |
| IF,SUM | 488,113 |
| COLUMNS,MID,REPT,SUBSTITUTE,TRIM | 459,785 |
| WEEKDAY | 429,244 |
| IF,IF,IF,IF,IF,IF,IF,IF | 369,177 |
| MONTH | 368,708 |
| LEFT | 363,611 |
| _xlfn.XLOOKUP | 343,869 |
| IF,ISBLANK | 336,105 |
| IF,IF,IF | 335,063 |

to standard zero-shot evaluation. We hypothesize that this outcome is because our tasks primarily require direct semantic mapping and precise formula generation, rather than the multi-step logical or numerical reasoning where CoT typically excels.

## 8.1 Accessing Sheetpedia

The Sheetpedia corpus and associated benchmarks are available for research purposes. Researchers interested in accessing the dataset or exploring the NL2SR and NL2Formula tasks can contact the authors or visit the project repository at https://huggingface.co/datasets/tianzl66/Sheetpedia and https://github.com/TTtianTT/Sheetpedia. We hope these supplementary materials facilitate further advancements in spreadsheet intelligence and NLP research.

Table 6: Top 30 Most Used Excel Formulas: After Deduplication

| Formula | Count |
| --- | --- |
| Plain Formula | 22,307,751 |
| IF | 4,666,924 |
| SUM | 3,101,131 |
| VLOOKUP | 1,672,540 |
| AND,IF | 1,317,541 |
| IF,IF | 938,261 |
| COUNTIF | 811,864 |
| IFERROR,VLOOKUP | 652,822 |
| IFERROR,__xludf.DUMMYFUNCTION | 613,323 |
| TEXT | 596,695 |
| IF,VLOOKUP | 446,741 |
| INDEX,MATCH | 429,405 |
| AVERAGE | 414,849 |
| IFERROR,INDEX,MATCH | 407,403 |
| COUNTIFS | 404,876 |
| IFERROR | 403,389 |
| SUMIFS | 388,684 |
| IF,IF,IF,IF | 302,367 |
| SUMIF | 284,157 |
| CONCATENATE | 282,101 |
| CHAR,CHAR,CLEAN,SUBSTITUTE,SUBSTITUTE,TRIM | 259,789 |
| COUNTIF,IF | 256,165 |
| IF,SUM | 243,636 |
| IF,ISBLANK | 242,460 |
| LEFT | 241,113 |
| IF,IF,IF,IF,IF,IF,IF,IF | 234,509 |
| COLUMNS,MID,REPT,SUBSTITUTE,TRIM | 224,934 |
| _xlfn.XLOOKUP | 222,150 |
| IF,IFERROR | 216,615 |
| MONTH | 214,274 |

Table 7: Performance comparison of Chain-of-Thought (CoT) prompting versus standard zero-shot prompting on NL2SR and NL2Formula tasks.

| Model | NL2SR (CoT) | NL2SR (0-shot) | NL2Formula (CoT) | NL2Formula (0-shot) |
| --- | --- | --- | --- | --- |
| GPT-4o | 67.56 | 66.67 | 50.00 | 63.33 |
| LLaMA-3.1-8B | 12.50 | 41.67 | 18.33 | 27.50 |
| LLaMA-3.3-70B | 54.17 | 66.67 | 28.57 | 47.50 |
| Qwen2.5-7B | 31.67 | 45.83 | 26.63 | 30.83 |
| Qwen2.5-72B | 73.57 | 75.83 | 56.03 | 60.00 |

## NL2SR Query Generation Prompt

System: Given the content of the spreadsheet and the selected cell
ranges, your task is to generate an accurate query that precisely
reflects the data within these ranges. The query should reveal the
semantics of the selected cell ranges. When the query is given, we
should be able to pinpoint a unique cell range in the sheet.
Importantly, this unique range should be identical to the selected
cell range. The generated query must solely reflect the data within
the range and should not include any other cells. While generating
an accurate query, you should aim to keep the query as concise as
possible. Avoid referring to the address and content of the
selected cell. Avoid lengthy queries. The generated query should
be in line with my needs as a user, as if it's a succinct request
made during a conversation with the model.

The sheet data will be provided to you in a format as follows: Each data
cell in the spreadsheet is represented by a pair consisting of the cell
address and cell content, separated by a comma, such as 'A1,Year'. This
means that 'A1' is the cell's address, and 'Year' is its content. Cells
are separated by a vertical bar ('|'), like 'A1,Year|A2,Profit'. The cell
content can be empty, resulting in cell data like 'A1,|A2,Profit'. Cells
are organized in row-major order, with different rows in the spreadsheet
separated by line breaks. If there are merged cells in the sheet, they
are split into multiple cells and only the first cell will be filled with
content, other cells will be left as blank. You can visualize the sheet
data as a matrix of cells. Following the matrix, all the merged cells are
provided in the format '<upper-left address>:<lower-right address>,' like
'A3:C3', with each line representing one merged cell.

Example User:

{Example i:}
Original Sheet Content:
{Example Sheet}

The selected cell ranges: {Example Range Info}

Example Assistant:

Corresponding Query: {Example Query}

User:
Now your turn,

My Sheet Content:
{Sheet String}

The selected cell ranges: {Range Info}

Output Format: Please generate a query based on the selected cell ranges
. The output should be provided in a JSON format, with a key of 'query'
and the generated query as the corresponding value.

Tell me the content in the selected cell ranges first and then generate
the corresponding query, keep the query precise and concise.

Figure 7: NL2SR Query Generation Prompt

## NL2Formula Query Generation Prompt

```
System: You will receive sheet data along with an Excel formula. Your
task is to generate a query that can reflect the semantics of the
corresponding formula.

The sheet data will be provided to you in a format as follows: Each data
cell in the spreadsheet is represented by a pair consisting of the cell
address and cell content, separated by a comma, such as 'A1,Year'. This
means that 'A1' is the cell's address, and 'Year' is its content. Cells
are separated by a vertical bar ('|'), like 'A1,Year|A2,Profit'. The
cell content can be empty, resulting in cell data like 'A1,|A2,Profit'.
Cells are organized in row-major order, with different rows in the
spreadsheet separated by line breaks. If there are merged cells in the
sheet, they are split into multiple cells and only the first cell will
be filled with content, other cells will be left as blank. You can
visualize the sheet data as a matrix of cells. Following the matrix,
all the merged cells are provided in the format '<upper-left address>:
<lower-right address>,' like 'A3:C3', with each line representing one
merged cell.

{Output Format}

{Guidelines}

Example User:

Original Sheet Content:

{Example Sheet}

{
    "formula": Example Formula,
    "address": Example Address
}

Example Assistant:

Corresponding Query: {Example Query}

User:
Now your turn,

{
    "sheetString": {Sheet String},

    "Formula Address": {

        "formula": {Formula},
        "address": {Address}

    }

}
```

Figure 8: NL2Formula dataset query generation prompt

**NL2SR Query Scoring Prompt**

```
[Task]
Evaluate the quality of this natural language query for describing an
Excel cell range. Consider:

1. Clarity (0-3): Is the purpose/use of the cell range unambiguous?
2. Accuracy (0-3): Does the query correctly reflect the cell range's
application (e.g., data processing, references)?
3. Conciseness (0-2): Is it free of unnecessary details?
4. Completeness (0-2): Are all the meanings of the cells included in the
query?

[Scoring Rubric]
• 9-10: Perfectly describes the cell range's scope and purpose
• 7-8: Minor inaccuracies or omissions
• 5-6: Partial accuracy with vague references
• <5: Fails to characterize the cell range

[Input]
Cell Range: {cell_range}
Context: {context}
Query: {query}

[Output Format]
Strict JSON:
{
    "score": total_score,
    "breakdown": {
        "clarity": score,
        "accuracy": score,
        "conciseness": score,
        "completeness": score
    },
    "rationale": "Brief explanation"
}

IMPORTANT:
1. Do NOT include any additional text before or after the JSON object
2. Ensure the JSON is valid and properly formatted
```

Figure 9: NL2SR Query Scoring Prompt

## NL2Formula Query Scoring Prompt

```
[Task]
Evaluate the quality of this natural language query for describing an
Excel formula. Consider:

1. Clarity (0-3): Is the calculation purpose unambiguous?
2. Accuracy (0-3): Does it match the formula's logic?
3. Conciseness (0-2): Is it free of redundant information?
4. Completeness (0-2): Are cell ranges/specifics included?

[Scoring Rubric]
• 9-10: Perfectly captures all formula aspects
• 7-8: Minor omissions but generally accurate
• 5-6: Partial accuracy with some ambiguities
• <5: Significant discrepancies

[Input]
Formula: {formula}
Context: {context}
Query: {query}

[Output Format]
Strict JSON:
{
    "score": total_score,
    "breakdown": {
        "clarity": score,
        "accuracy": score,
        "conciseness": score,
        "completeness": score
    },
    "rationale": "Brief explanation"
}

IMPORTANT:
1. Do NOT include any additional text before or after the JSON object
2. Ensure the JSON is valid and properly formatted
```

Figure 10: NL2Formula Query Scoring Prompt

**NL2SR Test Prompt**

```
[Task]
As a data scientist, you are presented with a spreadsheet and a user
query. Your task is to interpret the spreadsheet and identify the
specific cell range that corresponds to the given user query. Ensure
the identified range should be as precise as possible, and should not
include any other irrelevant cells.

The sheet data will be provided to you in a format as follows: Each data
cell in the spreadsheet is represented by a pair consisting of the cell
address and cell content, separated by a comma, such as 'A1,Year'. This
means that 'A1' is the cell's address, and 'Year' is its content. Cells
are separated by a vertical bar ('|'), like 'A1,Year|A2,Profit'. The
cell content can be empty, resulting in cell data like 'A1,|A2,Profit'.
Cells are organized in row-major order, with different rows in the
spreadsheet separated by line breaks. If there are merged cells in the
sheet, they are split into multiple cells and only the first cell will
be filled with content, other cells will be left as blank. You can
visualize the sheet data as a matrix of cells. Following the matrix,
all the merged cells are provided in the format '<upper-left address>:
<lower-right address>,' like 'A3:C3', with each line representing one
merged cell.

Output Format: Please generate the cell range based on the user query.
The output should be provided in a JSON format, enclosed in '''Json
and ''' markdown code blocks, with a key of 'cell range' and the
generated cell range as the corresponding value.
```

Figure 11: NL2SR Test Prompt

## NL2Formula Test Prompt

[Task]
As a data scientist, your task is to generate a formula based on a specific cell in a given spreadsheet. The query of the formula is provided. Use the spreadsheet and the query to generate the correct formula.

The sheet data will be provided to you in a format as follows: Each data cell in the spreadsheet is represented by a pair consisting of the cell address and cell content, separated by a comma, such as 'A1,Year'. This means that 'A1' is the cell's address, and 'Year' is its content. Cells are separated by a vertical bar ('|'), like 'A1,Year|A2,Profit'. The cell content can be empty, resulting in cell data like 'A1,|A2,Profit'. Cells are organized in row-major order, with different rows in the spreadsheet separated by line breaks. If there are merged cells in the sheet, they are split into multiple cells and only the first cell will be filled with content, other cells will be left as blank. You can visualize the sheet data as a matrix of cells. Following the matrix, all the merged cells are provided in the format '<upper-left address>: <lower-right address>,' like 'A3:C3', with each line representing one merged cell.

Output Format: Please generate the formula based on the user query. The output should be provided in a JSON format, enclosed in ```Json and ``` markdown code blocks, with a key of 'formula' and the generated formula as the corresponding value.

Figure 12: NL2Formula Test Prompt

