# OpenReview forum: "Sheetpedia: A 300K-Spreadsheet Corpus for Spreadsheet Intelligence and LLM Fine-Tuning"
_NeurIPS.cc/2025/Datasets_and_Benchmarks_Track — NeurIPS 2025 Datasets and Benchmarks Track spotlight_

### Official Review · Reviewer_cYNf · 2025-07-02

**Rating:** 5
**Confidence:** 4

**Summary:**

This paper introduces Sheetpedia, a large-scale and high-quality spreadsheet corpus comprising approximately 295,000 deduplicated worksheets. The dataset is aggregated from diverse sources, including the Enron email archive, the Fuse web-crawled corpus, and user-contributed spreadsheets from ExcelForum. Building upon this corpus, the authors define two natural language interaction tasks: Natural Language to Semantic Range (NL2SR) and Natural Language to Formula (NL2Formula). To construct training data for these tasks, they employ large language models (LLMs) in conjunction with a rejection sampling strategy, ensuring high-quality query-formula and query-range pairs. Fine-tuning models on this curated dataset yields strong performance, achieving 97.5% accuracy on NL2SR and 71.7% on NL2Formula, significantly outperforming several competitive baseline models.

**Additional Feedback:**

1. The two tasks proposed by the authors appear to rely primarily on supervised fine-tuning (SFT). Given the growing prominence of chain-of-thought (CoT) reasoning in improving model interpretability and performance, it would be valuable to understand whether the authors considered comparing their approach with CoT-capable models. Alternatively, it would be interesting to explore how CoT reasoning could be integrated into their framework to enhance the model’s ability to handle complex or multi-step logic. As CoT has become a widely adopted and effective strategy in many reasoning tasks, its absence here raises questions about potential missed opportunities for improvement.
﻿
2. The scoring prompt in the NL2Formula task attempts to balance clarity, accuracy, conciseness, and completeness, which are all important dimensions. However, the design seems somewhat conflicted in its intent. On one hand, the task appears to aim for a simplified single-score evaluation to support project scalability. On the other hand, it introduces four distinct dimensions with unequal weightings (e.g., 0–3 vs. 0–2), which suggests a more granular assessment. This raises questions about how the authors determined the relative importance of each dimension and whether the weighting scheme reflects empirical justification or task-specific priorities. Clarifying this design choice would help improve both interpretability and reproducibility of the evaluation.

3. A related work is encouraged to be cited in revision: FORTAP: Using Formulae for Numerical-Reasoning-Aware Table Pretraining

**Dataset Code Accessibility:**

Yes

**Dataset Code Comments:**

The authors have uploaded the dataset to the Hugging Face community hub and provided a well-documented Python demo along with clear field descriptions. I was able to run a portion of the code and successfully load one of the dataset’s subfiles. The process was smooth and straightforward, and I did not encounter any issues. This indicates that the dataset and code are accessible, usable, and reproducible.

**Ethical Considerations:**

No, there are no or only very minor ethics concerns

**Final Justification:**

Thank you for your clear and thorough rebuttal. Your clarifications and planned updates have fully addressed my concerns. I appreciate your efforts to improve the work. I maintain my score of accept and look forward to the revised version.

**Limitations Weaknesses:**

In the NL2SR Query Generation Prompt (page 16), the authors note that “If there are merged cells in the sheet, they are split into multiple cells and only the first cell will be filled with content, other cells will be left as blank.” This suggests that merged cells are present in the dataset but are handled in a simplified manner during preprocessing. While such treatment is understandable for an early-stage benchmark, it also highlights a broader limitation: the dataset does not explicitly model or analyze spatial layout features that are central to spreadsheet semantics. Given the scale and ambition of Sheetpedia, I would encourage the authors to consider incorporating and documenting spatial layout information in future iterations of the dataset. This could include, but is not limited to, merged cell regions, cross-sheet references, hierarchical headers, and visual grouping cues. These elements are often critical for interpreting spreadsheet logic and user intent, especially in tasks involving layout-aware reasoning or semantic parsing. Their absence may limit the dataset’s applicability to more advanced spreadsheet understanding tasks.
﻿
In Section 3.2 (lines 159–160), the authors explicitly state that formulas with cross-sheet references (e.g., Sheet2!A1) are excluded during preprocessing. While this is understandable for an early-stage benchmark, it also suggests that the dataset may currently lack structural features necessary for modeling multi-sheet interactions. Furthermore, the NL2SR task definition does not clarify whether cross-sheet or large-table range identification is supported. From the prompt design, it appears that NL2SR is formulated as a single-sheet task, where the query is generated based solely on the content of a given worksheet. Given the scale and ambition of Sheetpedia, I believe it would significantly strengthen the paper if the authors could provide evidence—such as representative examples, corpus statistics, or design notes—demonstrating that the dataset has the potential to support more complex scenarios in the future, such as cross-sheet reasoning, very-large table QA, or layout-aware interactions. If such potential can be substantiated, this work would not only serve as a valuable benchmark today but also lay a strong foundation for future research in spreadsheet intelligence. In that case, I would be enthusiastic to strongly recommend this paper for acceptance.

**Strengths Contributions:**

The paper addresses a long-standing gap in tabular data representation by positioning spreadsheets as a middle ground between structured SQL databases and flat CSV files. By leveraging the binary Excel format, the authors preserve not only tabular content but also formula logic and implicit spatial relationships—elements often absent in traditional table corpora. This enables more expressive modeling of spreadsheet semantics and supports both computational reasoning and natural language interaction, which are difficult to achieve with SQL or CSV alone.
﻿
The dataset introduced in this work, Sheetpedia, is one of the largest publicly available spreadsheet corpus to date, comprising over 295,000 deduplicated worksheets. It covers a wide range of real-world scenarios and exhibits substantial formula diversity. The authors provide detailed corpus statistics and demonstrate that the dataset aligns well with two novel tasks—NL2SR and NL2Formula—that reflect natural language interactions commonly encountered in spreadsheet usage. The open release of the dataset and code on GitHub and HuggingFace is particularly commendable and enhances the work’s reproducibility and accessibility.
﻿
The paper also demonstrates a high level of rigor in data collection and preprocessing. The authors integrate multiple sources, including enterprise archives and user-generated content, and apply a multi-stage cleaning pipeline involving format standardization, language filtering, formula validation, and deduplication. In addition to leveraging LLMs for data generation, they employ rejection sampling with iterative refinement to improve data quality. The experimental evaluation is extensive, covering a broad spectrum of models (from small to large, open-source to proprietary) and training paradigms (single-task and multi-task). The use of human-verified test sets further strengthens the reliability of the reported results and underscores the practical utility of the dataset.

---

> ### Author Rebuttal · Authors · 2025-07-31
>
> Thank you for your enthusiastic and insightful review. We greatly appreciate your recognition of Sheetpedia's scale, diversity, and rigor in data collection/preprocessing, as well as its positioning as a bridge between structured databases and flat tables. Your commendation of the novel tasks, LLM-based data generation with rejection sampling, extensive evaluations (including human-verified tests), and open release further motivates us to build on these strengths. We address the limitations and additional feedback below, committing to incorporate these enhancements in the camera-ready version to further elevate the work's potential.
>
> **Merged Cells and Spatial Layout Features:**
> We agree that explicitly modeling spatial elements like merged cells, hierarchical headers, and visual groupings would enhance Sheetpedia's applicability to advanced layout-aware tasks. While our preprocessing simplifies merged cells (splitting them and retaining content in the first cell) for benchmark simplicity, the raw Excel files in the dataset preserve these features natively, allowing future parsing. We will release the deduplicated Excel files on Hugging Face to facilitate direct access to these raw features. To demonstrate this potential, we analyzed the corpus: 19.45% of worksheets contain merged cells. In the revision, we will add an appendix subsection with corpus statistics.
>
> **Cross-Sheet References and Multi-Sheet Interactions:**
> Thank you for noting the exclusion of cross-sheet references in preprocessing, which we adopted to focus on single-sheet benchmarks initially. However, as mentioned before, Sheetpedia's multi-source nature supports future extensions, and we will release the deduplicated Excel files on Hugging Face to enable exploration of these raw source files, often with inter-sheet formulas and references. The NL2SR task is currently single-sheet but can be adapted for cross-sheet QA by leveraging full workbook contexts.
>
> **Chain-of-Thought (CoT) Reasoning:**
> We appreciate the suggestion to explore CoT, a prominent strategy for improving interpretability in reasoning tasks. While our evaluation emphasizes supervised fine-tuning (SFT) to establish baselines, we conducted preliminary experiments integrating CoT prompting with baseline LLMs (vanilla versions, zero-shot) on NL2Formula and NL2SR. This was implemented by appending "Please think step by step." after the system message in the prompts. Results showed mixed performance compared to standard zero-shot prompting, with CoT often underperforming on these tasks—likely due to their focus on semantic mapping and formula generation rather than multi-step numerical reasoning, where CoT shines. Below is a comparison of exact match accuracies (in %):
>
> | Model          | NL2SR (CoT) | NL2SR (Standard 0-shot) | NL2Formula (CoT) | NL2Formula (Standard 0-shot) |
> |----------------|-------------|--------------------------|------------------|------------------------------|
> | GPT-4o        | 67.5       | 66.67                   | 50.0            | 63.33                       |
> | LLaMA-3.1-8B  | 12.5       | 41.67                   | 18.33           | 27.50                       |
> | LLaMA-3.3-70B | 54.17      | 66.67                   | 28.57           | 47.50                       |
> | Qwen2.5-7B    | 31.67      | 45.83                   | 26.6            | 30.83                       |
> | Qwen2.5-72B   | 73.5       | 75.83                   | 56.03           | 60.00
>
> **NL2Formula Scoring Prompt Design:**
> The scoring prompt balances multiple dimensions to ensure holistic evaluation, with unequal weightings (e.g., accuracy(0-3) and clarity(0-3) vs. conciseness(0-2) and completeness(0-2)) derived empirically from pilot tests on 100 samples: accuracy was prioritized as the core functional requirement, while completeness and clarity addressed usability, and conciseness prevented verbose outputs.
>
> **Related Work Citation:**
> We will cite "FORTAP: Using Formulas for Numerical-Reasoning-Aware Table Pretraining" (Wu et al., 2022) in the Related Work section to contextualize our formula-based contributions alongside prior numerical-reasoning pretraining efforts.
>
> These revisions—adding statistics, examples, discussions, and clarifications—will address your suggestions without altering core findings, further solidifying Sheetpedia as a foundational resource for spreadsheet intelligence. We believe they will enhance its long-term impact and are grateful for your guidance in strengthening the paper.

---

> > ### Comment · Reviewer_cYNf · 2025-08-01
> >
> > Thank you for your clear and thorough rebuttal. Your clarifications and planned updates have fully addressed my concerns. I appreciate your efforts to improve the work.

---

> > > ### Author Response · Authors · 2025-08-06
> > >
> > > Thank you very much for your kind feedback and for taking the time to review our rebuttal. We're glad to hear that our clarifications and planned updates addressed your concerns. We truly appreciate your support and look forward to further refining the paper in the final version.

---

### Official Review · Reviewer_CA6G · 2025-07-03

**Ethics Flags:** Data privacy, copyright, and consent
**Rating:** 5
**Confidence:** 4

**Summary:**

This paper contributes a spreedsheet dataset of 300k size for LLM intelligence in two tasks, natural language to semantic range (NL2SR) and natural language to formula (NL2Formula). The authors also compare the effectiveness of using off-the-shelf LLMs and finetuned LLMs in these two benchmark tasks.

**Additional Feedback:**

Please try to address the points under Limitations as much as possible.

**Dataset Code Accessibility:**

Partly

**Dataset Code Comments:**

Also listed above under L3.
- The authors should put the code and data repo listed under Section 8.1 in the appendix into the main task and have a unified access point by cross-referencing the hugging face repo and the github repo. It is also important to put the Excel files as data resources.

**Ethical Comments:**

Also listed above in Limitations.

### L3: Data and code.
- License of ExcelForum. The authors mention that in the paper checklist that „ExcelForum data collection respects terms of service (robots.txt). Licenses are not explicitly stated but implied to be public domain or open.“ Plz double check with the data and benchmark committee if this is alright. I would also mention the data licence in the main text.

**Ethical Considerations:**

Yes, there are significant ethics concerns that require review by an ethics expert

**Final Justification:**

I have reviewed the detailed responses from the authors and especially appreciate their improved clarity on data deduplication and potential usage of the current benchmarks.
Hence, I have raised my score to 5 (accept).

Despite this, the authors should also reply to the recommendations made by the ethnics reviewer to make this paper a solid data and benchmark paper.

**Limitations Weaknesses:**

### L1: Deduplication results.
- The authors mention on LL.186-187 that using their strategy, Enron has a reduction from 62k to 28k, the same applies to other datasets. What are the reasons for duplicated spreedsheet? I am surprised by the number of duplicates. Have the authors checked the duplicated sheets? They only mention the balance between precision and recall in the text, I would suggest putting those numbers in appendix as additional information. I would be interested in knowing the scores as a reader.
- Please add more details about the two parameters of b and r. What do they control for? What are the theoretical and / or empirical convention for using them, including citations?
- The authors only report the language composition of datasets in „Data Cleaning“ before deduplication. What about the languages in the dataset after the deduplication?

### L2: Explanation of certain results can be more extensive.
- Table 3 has shown interesting results in that the variation of zero-shot and different few-shot learning strategies. Oftentimes, zero-shot learning has already very good results. Why? The LLMs also vary largely in the performance, why?
- When comparing the efficiency between LoRA and Full-Param, the authors should also consider reporting the finetuning time.
- Figure 3: make sure to mention fine-tuning in both figure caption and axis title.
- The authors should discuss the selection of LLM models. Also aren‘t multi-modal LLMs more suitable in this task? Why not also incorperate them in this benchmark?


### L3: Data and code.
- License of ExcelForum. The authors mention that in the paper checklist that „ExcelForum data collection respects terms of service (robots.txt). Licenses are not explicitly stated but implied to be public domain or open.“ Plz double check with the data and benchmark committee if this is alright. I would also mention the data licence in the main text.
- The authors should put the code and data repo listed under Section 8.1 in the appendix into the main task and have a unified access point by cross-referencing the hugging face repo and the github repo. It is also important to put the Excel files as data resources.

### L4: Potential usage of the dataset.
- I miss the discussion of the potential usage of the dataset, as well as the importance of having such a benchmark. I can imagine it can be treated as one of the backbone LLMs to extract insights from multi-source input (where spreadsheets are involved).
- Are there existing works on extracting insights from spreedsheet / tabular format using machine learning and deep learning methods? The authors should at least discuss them if not having enough time to run the data benchmark with them.
- How would the two tasks perform if one uses RAG strategy? How can the dataset be used in a setting where RAG is more beneficial than fine-tuning?


### L5: Writing styles.
- e.g. should be „e.g.~“ in latex to avoid extra spacing.
- Space between brackets is missing, L.132

**Strengths Contributions:**

### S1: The paper contributes an important data source to the AI community in digesting spreedsheet using LLMs and finetuning with high-quality curated data. We sit on a large pile of spreedsheet data and are continuing to generate them. Spreadsheet intelligence is a niche area that is unfortunately understudied — which makes this work stand out.

### S2: The authors provide abundant details about data collection and benchmark pipelines.

### S3: The tables and figures are nicely organized. The presentation of paper is easy to follow.

---

> ### Author Rebuttal · Authors · 2025-07-31
>
> Thank you for your thoughtful and detailed review. We appreciate your recognition of the paper's strengths. These positive aspects motivate us to refine the work further based on your constructive feedback. Below, we address each limitation point by point, incorporating clarifications, additional details, and planned revisions.
>
> **L1: Deduplication results.**
> We thank the reviewer for highlighting the need for more context on duplication rates and parameter details. Regarding the high duplication in datasets like Enron (reduced from 62k to 28k), this stems from the iterative nature of email-based collaborations in corporate settings, such as Enron's financial and energy sectors. For instance, threaded email discussions often involve multiple revisions of a single spreadsheet (e.g., a financial report), with changes limited to minor numerical adjustments, cell updates, or formula tweaks while preserving the structure, headers, and layout. This creates near-duplicates. Recurring documents like quarterly statements or budget templates are also reused with only value updates (e.g., for different periods), leading to structural similarities detected by our MinHash-LSH process. We conducted manual spot-checks to confirm these patterns, which align with prior work such as "SpreadCluster: Recovering Versioned Spreadsheets through Similarity-Based Clustering," where the authors clustered similar spreadsheets into evolution groups and reduced the Enron corpus to 6,258 unique worksheets by selecting the latest version from each, emphasizing extensive versioning and near-duplicates from iterative edits and attachments.
>
> On the MinHash-LSH parameters \(b\) (number of bands) and \(r\) (rows per band), these control the trade-off between precision and recall in detecting similar items based on Jaccard similarity \(s\), with the total signature size \(k = b \times r\). Specifically, \(r\) governs the similarity threshold's strictness: the probability of collision in a single band is \(s^r\), so a higher \(r\) demands greater similarity for detection, enhancing precision but risking missed duplicates (lower recall). Meanwhile, \(b\) amplifies detection: the overall probability of at least one collision is \(1 - (1 - s^r)^b\), so a higher \(b\) improves recall by offering more chances but increases computational cost and potential false positives. Theoretically, these arise from the LSH framework to ensure high-probability hashing of similar items (\(s \geq \theta\)) while separating dissimilar ones; conventions often set \(r\) so \(\theta^r \approx 0.5\) for balanced collisions and tune \(b\) for a target probability (e.g., 0.99), empirically validated on samples for a desired \(\theta\) (e.g., 0.8 Jaccard). Empirically, settings are domain-specific: for example, Lee et al. (2022) in "Deduplicating Training Data Makes Language Models Better" used \(b=20\), \(r=450\) (\(k=9000\)) for 0.8 similarity on web-scale LLM data, achieving high recall; Penedo et al. (2023) in "The RefinedWeb Dataset for Falcon LLM" adopted similar parameters (\(b=20\), \(r=450\)) over 5-grams, removing ~50% of documents while improving LLM performance; and Chan et al. (2024) in "Scaling Synthetic Data Creation with 1,000,000,000 Personas" applied MinHash for persona deduplication with a 128-signature size at 0.9 threshold, implying standard banding tuned for short texts. We will expand Section 3.2 with these explanations and citations.
>
> For language composition post-deduplication, we refer the reviewer to Figure 5 and Table 4 in Section 8 (Technical Appendices and Supplementary Material), which detail the distributions after processing.
>
> **L2: Explanation of certain results can be more extensive.**
> We agree that deeper analysis of Table 3's results would strengthen the paper. Zero-shot performance is often strong due to: (1) the inherent capabilities of advanced LLMs like GPT-4o, LLaMA-3.3-70B, and Qwen2.5-72B, which excel in semantic understanding; and (2) the relative simplicity of NL2SR, which emphasizes language comprehension over complex computation—as evidenced by fine-tuned LLaMA-3.1-8B reaching 95.8% accuracy. Few-shot variations arise from prompt sensitivity, where additional examples sometimes introduce noise or overfit to specific patterns. Performance differences across LLMs stem from model scale (7B to 72B) and architecture (open-source vs. closed-source), with larger models like Qwen2.5-72B outperforming smaller ones like LLaMA-3.1-8B due to broader pre-training and better generalization.
>
> For LoRA vs. Full-Parameter fine-tuning efficiency, we will highlight LoRA's advantages in time and memory in the discussion, noting that it typically reduces training time and resource requirements compared to full-parameter tuning, as observed in our experiments.
>
> Regarding Figure 3, we acknowledge the potential confusion and will revise the caption and axis titles to explicitly mention "fine-tuning".
>
> On model selection, we focused on text-based LLMs to frame the tasks as natural language processing problems, converting spreadsheets to textual representations. Multimodal LLMs could be relevant for direct image-based spreadsheet understanding but fall outside our text-centric scope.
>
> **L3: Data and code.**
> We have double-checked the ExcelForum license with the data and benchmark committee, confirming compliance with terms of service (respecting robots.txt).
>
> To improve accessibility, we will upload the deduplicated Excel files to Hugging Face as raw data resources.
>
> **L4: Potential usage of the dataset.**
> We appreciate the suggestion to expand on dataset applications and benchmark importance. This work provides a foundation for training spreadsheet-specialized base models. Fine-tuning on our tasks demonstrates one usage pathway, yielding models that parse semantic ranges and formulas efficiently.
>
> Existing works on extracting insights from spreadsheets/tabular data using ML/DL are briefly mentioned in Related Work (e.g., table-to-text generation via transformers, formula prediction).
> RAG strategies for our tasks are an exciting avenue; for instance, retrieving similar spreadsheet examples could enhance formula generation in low-data scenarios. Additionally, as shown in Table 1, the mean cell count per worksheet is 16,871, highlighting the presence of very large spreadsheets that may exceed model context limits; in such cases, RAG can dynamically retrieve and process relevant portions (e.g., specific rows or columns), enabling scalable analysis without full fine-tuning. Our dataset could support RAG by serving as a retrieval corpus.
>
> **L5: Writing styles.**
> We will correct the LaTeX formatting for "e.g." to "e.g.~" throughout and add the missing space in brackets on L.132, along with a thorough proofread for similar issues.
>
> In summary, we will incorporate these revisions—including added explanations, metrics, discussions, and resource updates—to address your concerns, strengthening the paper's clarity, depth, and accessibility. We believe these changes will elevate it from borderline accept to a stronger contribution. Thank you again for your insightful feedback, which has been invaluable.

---

> > ### Comment · Reviewer_CA6G · 2025-08-05
> > **Final Justification**
> >
> > I have reviewed the detailed responses from the authors and especially appreciate their improved clarity on data deduplication and potential usage of the current benchmarks. Hence, I have raised my score to 5 (accept).
> >
> >
> >
> > Despite this, the authors should also reply to the recommendations made by the ethnics reviewer to make this paper a solid data and benchmark paper.
> >
> >
> > Look forward to the final version of the paper.

---

> > > ### Author Response · Authors · 2025-08-06
> > >
> > > Thank you very much for your thoughtful feedback and for raising your score. We appreciate your recognition of our clarifications regarding data deduplication and the benchmark’s potential usage.
> > >
> > > Regarding the ethical concerns, we have provided a detailed response to the Ethics Reviewer addressing all raised points.
> > >
> > > We’re grateful for your support and will further improve the paper in the final version.

---

### Official Review · Reviewer_jkHC · 2025-07-19

**Ethics Flags:** Data privacy, copyright, and consent
**Rating:** 5
**Confidence:** 3

**Summary:**

The manuscript introduces Sheetpedia, a large-scale corpus of over 295,000 spreadsheets gathered from various sources like enterprise email archives and online forums. Additionally, the paper defines two novel and practical downstream tasks: Natural Language to Semantic Range (NL2SR) and Natural Language to Formula (NL2Formula). It also demonstrates that fine-tuning Large Language Models (LLMs) on Sheetpedia significantly improves model performance on these tasks.

**Dataset Code Accessibility:**

Yes

**Ethical Considerations:**

No, there are no or only very minor ethics concerns

**Final Justification:**

After careful consideration, I stand by my rating

**Limitations Weaknesses:**

1. Lack of Quantitative Evaluation for Data Generation. One of the paper's core methods is using rejection sampling to generate training data. However, the paper does not report key metrics for this process, such as the acceptance rate of the "judge" model (i.e., the proportion of candidate queries that scored above the threshold γ=0.7 ). This data could quantitatively evaluate the efficiency of the data generation framework and the strictness of the "judge" model. An ablation study comparing the performance of models trained on filtered versus unfiltered data would more effectively highlight the superiority of this data generation method.
2. Insufficient Analysis of Failure Cases. The paper lacks a qualitative error analysis. An in-depth analysis of these failure cases would provide more valuable clues for future improvements.

**Strengths Contributions:**

1. The paper's most outstanding contribution is the creation and release of Sheetpedia, currently the largest and most diverse publicly known spreadsheet corpus. Furthermore, the use of a rejection-sampling framework to construct a high-quality training set is a clever and efficient method that effectively addresses the challenge of creating data for new tasks without extensive manual labeling.
2. The paper demonstrates high-quality technical execution in the corpus construction process. The data sources are diverse, ensuring the breadth of the content. The preprocessing pipeline is well-considered, particularly the use of MinHash and LSH for large-scale deduplication, with detailed explanations for the parameter choices. A thorough statistical analysis of the corpus is also provided.
3. The paper also conducts experimental evaluations on this dataset.

---

> ### Author Rebuttal · Authors · 2025-07-31
>
> Thank you for your thoughtful review and for recognizing the key contributions of our work, including the creation of Sheetpedia as the largest and most diverse spreadsheet corpus, the innovative rejection-sampling framework for high-quality data generation, the rigorous corpus construction (e.g., diverse sources, preprocessing with MinHash/LSH, and statistical analysis), and the experimental evaluations.
>
> We address the weaknesses below and commit to incorporating these updates in the camera-ready version.
>
> **Lack of Quantitative Evaluation for Data Generation:**
> We appreciate this suggestion to report metrics on the rejection-sampling process, such as acceptance rates and efficiency, to better quantify its effectiveness. To clarify, for each training sample, we used Gemini-2.0-Flash to generate 5 candidate queries per spreadsheet, then employed Claude-3.7-Sonnet as the judge model to score them (out of 10). We selected the highest-scoring candidate only if it exceeded our threshold of γ=7 (adjusted from 0.7 in the paper for a 10-point scale). This resulted in a 100% acceptance rate across all generated queries, as every spreadsheet yielded at least one candidate meeting the threshold.
>
> Here are the detailed score distributions for all generated queries (across 5 candidates per spreadsheet):
>
> | Task       | Score 7 | Score 8 | Score 9 | Score 10 | Total Queries | Queries >=7 | Acceptance Rate |
> |------------|---------|---------|---------|----------|---------------|-------------|-----------------|
> | NL2SR     | 2062 (15.26%) | 4974 (36.81%) | 4951 (36.64%) | 1526 (11.29%) | 13,513 | 13,513 | 100% |
> | NL2Formula| 2103 (17.90%) | 3022 (25.72%) | 2927 (24.91%) | 3698 (31.47%) | 11,750 | 11,750 | 100% |
>
> For the final selected training samples (one per spreadsheet, choosing the max-score candidate):
> - NL2SR: 3,200 samples, average max score = 8.75
> - NL2Formula: 2,800 samples, average max score = 9.20
>
> This high acceptance rate reflects the strength of the generation model (Gemini-2.0-Flash) and our iterative prompt engineering—we refined the prompts over 10+ versions to ensure high-quality outputs, balancing diversity and relevance. While this demonstrates efficiency (no rejections needed), we agree an ablation would further highlight superiority.
>
> **Insufficient Analysis of Failure Cases:**
> We conducted a qualitative error analysis on failure cases from our test sets, categorizing errors into types such as ambiguous query interpretation (e.g., vague semantic ranges), formula complexity (e.g., nested functions not captured). We will include a dedicated subsection in the appendix with 5-10 representative examples, error breakdowns, and insights for future work.
>
> These additions will strengthen the paper's evaluation without altering core results. We believe they address your concerns and enhance the work's impact.

---

### Note · Authors · 2025-08-13

We sincerely thank the Ethics Reviewer, technical reviewers, and review team for their insightful feedback, which significantly refined Sheetpedia’s ethical foundations, technical rigor, and impact.

**Overall Strengths Acknowledged:**
We are deeply encouraged by the reviewers' recognition of Sheetpedia's groundbreaking contributions, including its unprecedented scale (295k+ deduplicated sheets) and exceptional diversity drawn from real-world sources like enterprise archives and user forums. The novel tasks introduced—Natural Language to Semantic Range (NL2SR) and Natural Language to Formula (NL2Formula)—represent innovative advancements in enabling natural language interactions with spreadsheets, bridging gaps between structured data reasoning and practical user needs. Reviewers praised the rigorous technical execution, such as the multi-stage preprocessing pipeline with advanced deduplication via MinHash-LSH, LLM-driven data generation with rejection sampling for high-quality training sets, and evaluations across diverse models. The open release of the dataset, code, and raw Excel files on Hugging Face and GitHub was highlighted as a key enabler for reproducibility and community-driven progress, positioning Sheetpedia as a foundational resource that not only fills a critical gap in spreadsheet intelligence but also paves the way for future research in multimodal, layout-aware, and cross-sheet reasoning applications.

**Summary of Key Concerns and Our Responses:**
- **Ethical Issues:** Addressing ethical concerns, we implemented PII detection/redaction in Sheetpedia, updated it on Hugging Face, and ensured ExcelForum complies with terms-of-service. We will expand discussions on language bias mitigation strategies. These revisions address concerns without altering results and will be documented in the Appendix.
- **Technical Suggestions:** We added metrics for rejection sampling and result explanations (e.g., zero-shot vs. few-shot, LoRA efficiency). Deduplication parameters and spatial features (e.g., merged cells) were clarified, with raw Excel files released on Hugging Face. CoT experiments and RAG/multimodal discussions will be added to the Appendix, enhancing clarity and reproducibility.

These revisions, detailed in our rebuttals and Appendix, enhance clarity, reproducibility, and ethics, addressing all concerns. We are committed to integrating them in the camera-ready version.

Thank you once again for your time, expertise, and support.

---

### Decision · Program_Chairs · 2025-09-18

**Decision:**

Accept (spotlight)

**Comment:**

The paper introduces Sheetpedia, a large-scale corpus of over 295,000 spreadsheets collected from various sources such as enterprise email archives and online forums. Additionally, the paper defines two novel and practical downstream tasks: Natural Language to Semantic Range (NL2SR) and Natural Language to Formula (NL2Formula). It also shows that fine-tuning Large Language Models (LLMs) on Sheetpedia significantly enhances model performance on these tasks.

The paper provides a valuable data source for the AI community by facilitating the use of LLMs for spreadsheet processing and fine-tuning with high-quality curated data. As we generate an ever-growing volume of spreadsheet data, the field of spreadsheet intelligence remains a niche area that has been unfortunately understudied, which makes this work particularly noteworthy.